# *“It Is Like We Are Living in a Different World”*: Health Inequity in Communities Surrounding Industrial Mining Sites in Burkina Faso, Mozambique, and Tanzania

**DOI:** 10.3390/ijerph182111015

**Published:** 2021-10-20

**Authors:** Andrea Leuenberger, Olga Cambaco, Hyacinthe R. Zabré, Isaac Lyatuu, Jürg Utzinger, Khátia Munguambe, Sonja Merten, Mirko S. Winkler

**Affiliations:** 1Swiss Tropical and Public Health Institute, P.O. Box, CH-4002 Basel, Switzerland; olga.cambaco@manhica.net (O.C.); zraogohyacinthe@yahoo.fr (H.R.Z.); ilyatuu@ihi.or.tz (I.L.); juerg.utzinger@swisstph.ch (J.U.); sonja.merten@swisstph.ch (S.M.); mirko.winkler@swisstph.ch (M.S.W.); 2University of Basel, P.O. Box, CH-4003 Basel, Switzerland; 3Manhiça Health Research Centre, Maputo C.P. 1929, Mozambique; khatia.munguambe@manhica.net; 4Research Institute of Health Sciences, Ouagadougou B.P. 7192, Burkina Faso; 5Ifakara Health Institute, P.O. Box, Dar es Salaam 78 373, Tanzania; 6Faculty of Medicine, Eduardo Mondlane University, Maputo C.P. 257, Mozambique

**Keywords:** community-based research, equity, extractive industries, focus group discussion, health impact assessment, social determinants of health, sub-Saharan Africa, Sustainable Development Goals

## Abstract

Background: Health equity features prominently in the 2030 Agenda for Sustainable Development, yet there are wide disparities in health between and within countries. In settings of natural resource extraction (e.g., industrial mines), the health of surrounding communities is affected through myriad changes in the physical, social, and economic environment. How changes triggered by such projects translate into health inequities is poorly understood. Methods: This qualitative study explores potential layers of inequities by systematically coding perceived inequities of affected communities. Drawing on the framework method, we thematically analyzed data from 83 focus group discussions, which enrolled 791 participants from 10 study sites in Burkina Faso, Mozambique, and Tanzania. Results: Participants perceived inequities related to their individual characteristics, intermediate factors acting on the community level, and structural conditions. Due to environmental pollution and land loss, participants were concerned about unsecured livelihoods. Positive impacts, such as job opportunities at the mine, remained scarce for local communities and were claimed not to be equally distributed among community members. Conclusion: Extractive industries bear considerable risks to widen existing health gaps. In order to create equal opportunities among affected populations, the wider determinants of health must be considered more explicitly in the licensing process of resource extraction projects.

## 1. Introduction

Many low- and middle-income countries (LMICs) are rich in natural resources, which encompasses both opportunities and risks for sustainable development. Indeed, natural resource extraction projects act on several of the Sustainable Development Goals (SDGs), including various health-related targets [1,2]. At the global level, it has been emphasized that extractive industries will contribute to achieving the 2030 Agenda for Sustainable Development [1,3,4]. In the strive toward a low-carbon future, the demand for metals and minerals is rising [5]. At the national level, sectors engaged in natural resource extraction (e.g., mining, oil, and gas) are important partners for economic development [6]. Fiscal revenues and public–private partnerships hold promise to improve public services, such as education and health care [7,8]. At the local level, industrial mining companies are becoming more and more engaged in community development through corporate social responsibility (CSR) programs, including health-related interventions [9,10].

Prospects for development that go hand-in-hand with the implementation and operation of natural resource extraction projects are opposed by potential negative consequences for health and wellbeing in surrounding communities. Affected communities face particular challenges related to environmental pollution, social disruption, or increased poverty [11,12,13]. Consequently, in communities living next to mining areas, an increased burden of diseases and poor wellbeing has been reported [14,15]. Due to conflicting interests of mining companies and local stakeholders, local communities might not benefit as much as reported [16]. Further, CSR has been described as an elusive concept, with varying perceptions of the effectiveness of CSR among different stakeholders [17]. Against this background, the question arises whether profit organizations are undermining health equity [8].

Creating equal opportunities to achieve good health and wellbeing is a core element of the 2030 Agenda for Sustainable Development [18,19]. To reduce health inequities and act toward a more equitable world, there is a pressing need to address the root causes [3,20,21]. Particularly in sub-Saharan Africa, where many countries are lagging behind in terms of health-related targets of the SDGs, acting on the social determinants of health is closely linked to poverty reduction [22,23].

The social determinants of health are defined as the conditions in which people are born, grow, live, and work, which has been conceptualized in various models [24,25]. Most models consider multiple layers, including individual and intermediate factors and structural conditions, which all affect people’s health and wellbeing [26]. Social exclusion and intersectoral action have been identified as key barriers hampering health equity [24,27]. Despite the increasing recognition of social determinants in global development, societal-level factors, such as power, among other structural drivers of inequities, tend to be omitted [28,29]. Especially in settings of industrial mining, which are often located in remote areas, these dynamics are major drivers for the social fabric of local communities [30]. Clearly, there is a need to ensure equal opportunities in order to achieve good health and wellbeing in mining settings and toward sustainable development, while paying particular attention to the most marginalized population groups [31,32]. Yet, pathways of health impacts and related health inequities induced by natural resource extraction projects are poorly understood. Previous studies have been investigating specific factors that are closely linked to health and equity [33,34,35]. However, studies addressing health comprehensively, including its underlying causes, remain scarce in the context of natural resource extraction in LMICs. To address this issue, particularly qualitative studies that incorporate the voices of affected communities, are needed to deepen the understanding of local perceptions and beliefs of health and wellbeing [36,37,38]. Based on the perception of affected communities, the underlying values of health inequity can be made more explicit [39,40]. Qualitative research holds promise to advance equity [41], which is an essential feature for the achievement of the 2030 Agenda for Sustainable Development [42].

The overarching goal of this study was to deepen the understanding of health equity in communities living in close proximity to industrial mining projects. We addressed the following research questions: (i) What are perceived inequities induced by industrial mining projects in sub-Saharan Africa? (ii) How do the perceived inequities relate to different wider determinants of health? (iii) How do perceived inequities translate into health inequities?

## 2. Materials and Methods

### 2.1. Study Set-Up

This paper is embedded in a large research project pertaining to health impact assessment (HIA) of natural resource development and management [2,43]. Within this frame, a qualitative study was conducted in rural communities surrounding industrial mining sites [43]. The current piece includes data from study sites in Burkina Faso, Mozambique, and Tanzania, which are all partner countries of the framing research project and extracting natural resources for several decades. Figure 1 shows the study sites. A detailed description of the overall research project and the individual study sites is available elsewhere [14,43].

### 2.2. Ethical Approval

Ethical approval was obtained from national and institutional review boards in the three project countries and in Switzerland. In brief, the study was approved by the Ethics Committee for Health Sciences in Burkina Faso (no. 2019-013), the Institutional Committee on Bioethics in Health at Manhiça Health Research Centre (CISM) in Mozambique (no. CIBS-CISM/048/2018), the Institutional Review Board of the Ifakara Health Institute (IHI) (no. 32-2018) and the National Institute for Medical Research (NIMR) in Tanzania (no. 2969), and the Ethics Committee of Northwestern and Central Switzerland (EKNZ) (no. 2018-00386).

### 2.3. Recruitment and Study Population

Under the guidance of a local study coordinator (i.e., local health care professional or government officer), a transect walk was conducted in each study site at the beginning of the field work. This allowed identification of communities that are positively (e.g., community development initiatives) and negatively (e.g., environmental degradation or social disruption) impacted by the mine and subsequent recruitment of study participants. With the assistance of community leaders, adult community members (aged ≥18 years) were invited to participate in the study. Preferably, people who were familiar with the community and its dynamics in relation to the development of the mine were selected. All participants were informed about the purpose and procedures of the study and provided written informed consent prior to data collection. Participants were reimbursed for travel expenses or provided with snacks and refreshments according to local research standards and requirements.

### 2.4. Data Collection and Analysis

Focus group discussions (FGDs) were moderated by trained research assistants in the local languages. Sessions were held in gender-separated groups to minimize gender-based power relations that might impede participants to talk freely. Using a participatory approach, participants listed, categorized, and ranked perceived impacts of mining on community health. The individual part of the participatory approach facilitated engaged discussions with every participant, while particularly integrating also rather shy participants. All FGDs were audio-recorded and transcribed verbatim into French (Burkina Faso), Portuguese (Mozambique), or English (Tanzania).

Data analysis was done by two researchers, namely A.L. (English and French transcripts) and O.C. (English and Portuguese transcripts) employing Nvivo (Nvivo 12 Pro, QSR International; Melbourne, Australia). Therefore, an overarching “inequity” code was utilized to capture perceived inequities in an initial step, which was subsequently analyzed in depth for emerging themes. Specifically, when participants opposed positive attributes to negative consequences in the same phrase or participants compared themselves to others getting better or worse in response to the developments induced by the extractive industry, this was coded as inequity. For example: “*They have water in the mining, the water ends there at the fence, why is it impossible to bring it to us here?*”

For in-depth analysis of the inequity data, we created a specific coding system based on the wider determinants of health [44]. In brief, all statements in the initially created and overarching “inequity” node were re-coded by drawing on the framework method for qualitative data analysis [45]. The different layers of the health determinants were utilized as initial categories for the coding and complemented with emerging themes. An overview of the different categories, along with indicative quotes, is given as Appendix A (Table A1).

The original “rainbow” model of the social determinants of health consists of five basic layers, namely (i) personal factors; (ii) individual lifestyle factors; (iii) social and community networks; (iv) living and working conditions; and (v) general socioeconomic, cultural, and environmental conditions. Grounded on the same layers, a more recent model of the World Health Organization (WHO) emphasizes different topics of the living and working conditions and specifically link them to the SDGs [46]. In the current study, we applied the basic layers of original model of Dahlgren and Whitehead [44] as analytical device and to situate our findings.

## 3. Results

### 3.1. Study Sites and Participants

The study was conducted in communities around nine industrial mining sites in Burkina Faso, Mozambique, and Tanzania (Figure 1). The core resources extracted in these mining sites are gold (in six sites), and coal, ruby, and titanium (in one site each). Except for one study site in Mozambique, all mines have been operating for several years. In total, 83 FGDs were conducted with 791 participants (406 females, 385 males), who have been living in the impacted communities since, on average, more than 25 years. The characteristics of study participants are summarized in Table 1. The typical participants had less than four years of formal education. Participants were mostly farmers or running their own small businesses. Of note, artisanal mining was also reported as important income-generating activity during the discussions.

### 3.2. Perceived Inequities

Perceived inequities were linked to a broad range of factors, which are closely interrelated. The complexity of the perceived inequities are illustrated by the following quote:

“*Since [the mine] came to [our region] here, they have brought many problems to the people. By taking our land, they have brought many palaver between us. They have flatten us, as they have taken us to the mine and only employed us for six months and then left us again. Yet, you didn’t have your fields for cultivation anymore. Really, this is what causes problems. It can lead to crimes and thefts.*”(BF3_L1)

Despite this intersection of social, environmental, and economic changes, we present our findings in the following section layer by layer as adapted based on our findings. By referring to specific examples from the FGDs, we describe the perceived inequities, how these are interpreted as health inequities, and how they are located in the different layers of health determinants.

#### 3.2.1. Personal Factors: Place of Origin or Residence, Gender, and Age

During the FGDs, people’s place of origin or residence was linked most often to perceived inequities. For example, they complained that autochthonous people (“*natives*” or “*locals*”) fail to get employed in the mine. One participant said:

“*People living near the mine are not getting permanent employment opportunities, which have high salary. Getting high salary could enable us to provide for our families instead people coming from other regions are the ones getting good employment posts.*”(TZ1_L5)

In contrast, they often referred to “white people”, as workforce in the mine or potentially also investors of the mining projects. In several statements, the participants linked the activities undertaken by or the presence of white people to different problems, which affects the communities’ health and wellbeing. For example, one participant said:

“*This community is not healthy due to the works of these white people. […] Due to their activities, their blasting, affects us; we get different kinds of diseases.*”(TZ2_L8)

In Emakuwa (the local language in the northern part of Mozambique), “*the white [foreign person]*” also means boss. People refer to a person with more power and money as a white man, which does not necessarily mean the skin color. This perception is in line with the concept of “white-collar workers”, indicating the clear distinction to manual workers in industrial capitalism [47].

Gender and age were other personal factors that emerged related to perceived inequities. Participants’ statements suggested that females are disproportionally affected due to their subordinated position in the domestic and work environment. Moreover, males were more likely to benefit from the mining operations, particularly in terms of job opportunities. Impacts on men’s and women’s engendered roles and implications on gender and health equity are discussed in more detail elsewhere [48]. During the FGDs different age groups were mentioned. It was emphasized that children and adolescents are particularly vulnerable to the impacts of the mine and related consequences for their health.

Overall, reported personal factors were mentioned most often in relation to restricted opportunities for local residents to work in the mines and the different susceptibility for diseases of different population groups. Beyond individual biological factors (i.e., sex and age), place of origin or residence were also reported as important reasons for being advantaged or disadvantaged in the given context. Places of origin or residence were perceived as key characteristics, especially related to the influx of people from different regions, countries, and continents as well as the introduction of formal job opportunities in remote areas.

#### 3.2.2. Personal Resources

This category embraced individual, socially determined factors, including educational attainment and socioeconomic status. Education was a key issue as community members felt left behind due to their low education (Table 1). Having attended few years of formal education was often perceived as disadvantage compared to formal employees of the mines. Hence, the community members’ educational background was reported as the main reason for not getting employment in the mine. This is illustrated in the following quote:

“*Neither for jobs nor for anything they say that jobs already have owners […] the ones who know how to write and they used say that ‘you don’t know how to write’, but a long time ago they moralized us with jobs.*”(MZ2.1_L6)

Personal monetary resources and particularly the lack or depletion thereof, was a major concern among participants in relation to the development of the mining projects. The following quote underscores this point:

“*We are very poor now it is because of these whites. In the past we were not that poor.*”(MZ1_L4)

Community members explained that their financial situation worsened since the mine started to operate, due to the loss of their agricultural land or artisanal mining areas, which used to be reliable sources of income. Participants also mentioned that they are spending money increasingly for various goods or services such as food or health care. They asserted that since the operation of the mine, they are not able to produce food themselves anymore and they need to seek health care more often. In Tanzania and Mozambique, money was also needed to obtain a certain job position, such as a local security guard of the mine. Nevertheless, participants from Tanzania also reported about the support from the mine granted to the community fund to realize community projects, such as construction of new water systems or strengthening of health facilities. These contributions were particularly acknowledged, as they could use their own money for personal needs, including food and health care. Cash compensation for being resettled were short windows of opportunities, but respondents were not able to benefit in the longer term.

Despite potential benefits, study participants were most concerned about fulfilling their personal needs and the needs of their family based on their limited resources. This was reported to be exacerbated by the implementation of the mines. The “lifestyle” of participants was marked by surviving with limited resources, which contrast to lifestyle factors based on free choices as indicated by Dahlgren and Whitehead [44]. Participants reported to be particularly constrained in terms of their financial resources due to the implementation of the mine. Except for rare occasions to become employed, individuals were hardly able to benefit from the industrial mining projects. These findings indicate the aggravation of their poor economic conditions.

#### 3.2.3. Social and Community Network

During the FGDs, inequities related to social and community network were closely linked to the social disruption of local communities. Participants primarily reported that not everyone is equally benefitting from potential positive impacts. For example, access to interventions implemented in distant communities, job opportunities offered or the support of selected local associations or community-based organizations were not equally distributed among communities and community members. According to their statements, the unequal distribution caused tensions or even conflicts among the community members, which were absent before implementation of the mines. In several study sites, community members perceived that certain community leaders are benefiting disproportionally. Especially those who were selected by the mining companies to become liaison officers between the mine and the community. Participants linked this increase in power relations also to physical and material benefits, such as houses or vehicles for leaders. Moreover, power exacerbation through illicit charging or other forms of corruption was reported as recurrent theme related to the unequal distribution of benefits. Taken together, the social cohesion among the community declined as expressed by a participant from Mozambique:

“*The government is the one who causes the struggle, for you to be community leader you need to pay someone, now people are fighting to be community leader, those who had no decent house have built it […]. Those who never had a car now have a car, and so they are fighting to be community representatives, nowadays people have already opened their eyes, no one is robbed only the farmers who go to the fields all the time.*”(MZ2.1_L4)

Beside community internal dynamics, participants also reported the poor relationship between them and the mining companies. Community members claimed empty promises, such as job opportunities or wealth, or were not satisfied with the reallocation compensation payment and felt that their problems were ignored. This was perceived to be particularly frustrating for the local communities, as the mining companies were located on and benefiting from their land. In-migration was also reported to affect the social dynamics negatively and was perceived to increase the risk of infection with human immunodeficiency virus (HIV) through commercial sex work. While participants mentioned positive aspects related to in-migration, such as the possibility to generate an income through sexual transactions, negative aspects prevailed. Participants mentioned explicitly that they are more likely to be HIV positive because of foreign people working in the mine, with women and adolescent girls being particularly affected.

Social and community networks emerged as a central aspect of inequities, which is consistent with the concept introduced by Dahlgren and Whitehead [44]. In the setting of industrial mining, the rapid influx of people, the gain of power of selected community leaders, and the unequal distribution of benefits particularly influenced the social dynamics in affected communities, which had negative consequences for their health and wellbeing.

#### 3.2.4. Living Environment and Subsistence Work

Changes in the natural environment, which were induced by the mining projects, heavily affected the daily lives and work of communities and certain inequities emerged, including issues related to land, water, health care, roads, and electricity. Most importantly, the loss of land was a particular issue for subsistence farming, meaning the agricultural production to eat as well as sell goods. Participants reported that their access to land (for agriculture, herding, or housing), water (for fishing), or mining areas (for artisanal mining) was restricted by the mining companies. Despite being resettled or compensated for the land, participants were not satisfied as new areas were smaller than expected and the soil reported as infertile. Hence, creating an income in their traditional way was a struggle for communities surrounding industrial mining sites. Regarding their housing situation, participants mentioned that they did not only lose their land, but also their rights to possess the land. In Burkina Faso and Tanzania, they reported that the mines are owning large areas beyond the actual fence line, including the living areas of some communities. Consequently, they were not able to buy new land for housing and felt disenfranchised as their land certificates were of no value as many participants reported. Because of the explosions in the mine, many houses were reported to be cracked, and people expressed a need to move or construct new houses. Participants felt disentitled, as they were constrained to construct new buildings but also no longer allowed to dig holes for latrines or to bury their close relatives. Particularly in the dynamic setting of resource extraction, land-use conflicts, and the insecure housing situation are major issues, also for their health and wellbeing, as illustrated in the following quote:

“*The areas that everyone of us has exploited, were our property. In contrast, the area of the mine, which we are occupying today, is the property of these “white” [from the mine]. Why [?] Because the certificate for residential area as promised by the responsible from the mine, we did not receive it […]. Because you are not the owner of something, you are always living with fear. This problem affects our sleep.*”(BF1_L5)

Respondents complained about polluted water sources and reduced water availability due to the extraction activities. If respondents observed mining employees drinking bottled water, this further enhanced their perception. New water sources were installed by the mining companies to make up for the problem. However, participants’ needs for clean water were not met, as new water access points were reported to be crowded or located far away. With regard to their health, FGD participants noted an increase in water-borne diseases.

Similarly, inequities related to food production emerged. Agricultural fields and crops were reported to be polluted by toxic dust or water released by the mine or not growing on the polluted soil. Hence, food insecurity and hunger related to the implementation of the mines were an issue raised during the FGDs. These statements were opposed by an observation from a participant in Burkina Faso that mining employees get served a lot of meat.

Another topic discussed was health care services and thus, directly related to health equity. Accessibility and affordability of health care were key concerns among community members from all sites. Newly constructed or improved health care facilities through the support of the mining companies were acknowledged as a positive development by the participants. At the same time, they also felt that they had to seek health care more often because of the various health impacts induced by the mining operations. Some participants felt deceived by the mining companies:

“*The presence of dispensary is not for the intention of saving our lives but to destroy us because if it wouldn’t have been their mining activities, we wouldn’t have been getting sick frequently.*”(TZ3_L4)

Some inequities were also perceived in relation to the road network. Indeed, accessibility of the mine is key for the development of the mine and for transporting extracted material. Hence, improved roads were mentioned to be helpful for the mobility communities, especially with regard to reaching health facilities in due time. However, many more remote villages were not benefitting from these infrastructure developments. In Mozambique, for example, participants reported that new roads and bridges were not constructed as promised at the beginning of project development.

Another difference between people living in the community and the mines was access to the power grid. Participants in Tanzania mentioned a lack of electricity, while witnessing the power line for the mine passing their village. The same community was located next to the mining site, which was surrounded with an electric fence (personal observation). Respondents stated that electricity could be particularly helpful to have night-lights as an intervention to increase safety in their villages by reducing crimes and robberies.

Taken together, communities revealed inequities by comparing their basic living standards in and with the natural and physical environment with the modernity and prosperity of mining employees. They lost their natural resources for living and, hence, also their source of income to pay for basic needs, and thus their subsistence farming. Despite positive contributions of the mines to public infrastructure, negative perceptions prevailed and decreased the health and wellbeing of affected communities. Adapted to the communities’ situation, we comprised various aspects of the daily life and self-sustained work in this layer. Distinct from their living conditions and subsistence work, respondents reported about job opportunities in the mines.

#### 3.2.5. Job Opportunities and Working Conditions

Based on the FGDs, the theme of job opportunities and working conditions for local residents in the mine emerged as a major theme of perceived inequities. With the construction and operation of the mining projects, participants were—based on the announcement of mining companies—hoping for new job opportunities and having a regular income. However, unemployment was a major concern and participants expressed their frustration about unkept promises, not been qualified for jobs offered, and a limited number of jobs. Consequently, many participants struggled with generating an income despite the project development, as they reported:

“*They are employing chef who gets high salary while that job can be done by one of us from this community. When they were introducing the mining company they said natives will benefit a lot from the mining but we are only getting temporary employment for two weeks or two months or three.*”(TZ2_L1)

Available job opportunities were related to harsh working conditions, low salaries, or socially unprotected employment. The few job opportunities, such as local security guards, ended usually after a few months. Although participants acknowledged this source of income, they were disappointed that they did not benefit from certain standards provided for formal employees of the mining companies, such as high salaries, health care, and decent housing. Having an income is important for the communities to be able to afford basic needs such as food, education, and health care services.

Although “work environment and unemployment” are included in the original model, we used a separate layer for “job opportunities and working conditions in the mine”. The separate layer indicates that the concept of employment for local communities was rather new, as they traditionally are subsistence farmers or entrepreneurs (Table 1). Notably, a separate layer illustrates the widening of inequities related to positive and negative aspects coming with these works. Moreover, it demonstrates the key role of the inclusion of local communities in economic activities to foster sustainable development while reducing inequities [49].

#### 3.2.6. General Socioeconomic Conditions and Political Context

The general socioeconomic conditions and political context were referred to by statements about the role and (expected) responsibilities of the government. General socioeconomic conditions included statement about the governments’ management of economic benefits of the mines or investment to community development. In Mozambique, participants explicitly mentioned the lack of investment for safeguarding people’s health and wellbeing. Participants criticized that revenues from the mining companies were not re-invested to benefit affected communities, as promised prior the project development. General political context referred to the relationship of the state with the multi-national mining companies or related regulations and laws. Participants in Burkina Faso and Mozambique felt unsupported by the government regarding their health. This was particularly frustrating for participants, after having voted for the government, as they narrated:

“*In any case, the authorities must know that their power come from the people and without the people there is no power. In this regard, the government has the obligation to surveil the health of the population.*”(BF1_L5)

In Tanzania, as in Burkina Faso, several community development projects were implemented and acknowledged by the respondents in relation to a law. The regulations require that a certain share of the companies’ revenue must be invested in community development projects.

Despite the important role of mining companies for the national economies, as repeatedly recognized by the participants, their statements indicate the lack of sustained improvements in affected communities, including health and wellbeing. In our context, general conditions were mostly shaped by the role of the governments and national regulations. Socioeconomic or political conditions were revealed, indicating the challenge to manage multi-national partnerships and ensure benefits also on the local level.

### 3.3. Consistency of Findings across Countries

As shown in Figure 2, similar patterns of inequities were observed in the three study countries. In all sites, perceived inequities related to job opportunities and working conditions in the mine as well as living environment and subsistence work were mentioned most frequently. Perceived inequities related to personal factors (e.g., place of residence or origin, gender, and age), personal resources (educational attainment and financial resources), social and community network, and general conditions were revealed as secondary themes in all countries.

## 4. Discussion

We conjecture that natural resource extraction will play a role in sustainable development with health equity being a critical issue. This paper sought to deepen the understanding of health inequities in mining regions in different parts of sub-Saharan Africa, by exploring the perception of affected communities. The study revealed that operation of the mining projects, brought to light several inequities, as expressed in various statements of local communities highlighting “different worlds”. Based on the perception of local community members, our findings indicate that there is a clash of rural, economically, and socially disadvantaged communities living in close proximity to extractive industries in LMICs. Both positive and negative changes induced by the mines resulted in increased perceived inequities. Positive changes related to interventions or job opportunities that were not distributed equally among and within communities. Negative changes unsecured the livelihoods of communities investigated and, hence, decreased their socioeconomic status. Despite improved health care services, communities’ opportunities to achieve good health and wellbeing were reported to be impeded by the construction and operation of the mine. Hence, this community-centred study indicates that the gap in health equity is widening in a highly dynamic and complex setting of industrial mining projects.

### 4.1. Complexity of Health Inequities

Many of the specific issues faced by the communities in relation to health equity have been studied before, including community dynamics [50,51], resettlement [52], environmental degradation [53], water infrastructure [54], land-use conflicts [55], and poverty [12]. Based on the emic perspective, our findings revealed that the participants perceive the situation as a complex interplay of different factors and that different layers were closely intersecting with each other. Especially due to the multi-dimensionality of the perceived inequities, communities reported them as particular challenges for their health. In the three study countries, respondents spoke about how the operation of the mines were responsible for social and environmental problems leading to an increased burden of diseases and ill health in the local communities. Being at the same time impacted by economic dynamics, affected communities who have been living in remote areas could not afford health services, let alone the transportation costs to reach them. Ultimately, the combination of different factors extracted from the individual layers draw a picture of unequal opportunities for health and wellbeing and, hence, show that there are important health inequities [56].

### 4.2. Locating Our Findings in a Model of Health Determinants

By referring to statements describing “different worlds” and the reported changes induced by the mining operations, we located the perceived inequities in the layers of wider determinants of health. As shown in Figure 3, the basic layers as suggest by Dahlgren and Whitehead [44], were useful to position the perception of local communities. However, our analysis pertained to perceived inequities revealed specific themes and sub-themes different from the original layers. For example, regarding personal factors in the central layer, place of origin or residence were (besides gender and age) important underlying factors for affected communities to feel excluded (e.g., job opportunity). The second layer embraces personal resources (including their educational background and financial resources), which is in contrast to the “lifestyle factors” proposed in the original framework. The participants’ way of living was mostly determined by their limited resources, which were reported to become aggravated by the changes induced through the construction and operation of the mine. Social dynamics within the communities were influenced by the mines through the unequal distribution of benefits and gains and exploitation of power relations of some community members. Hence, additional factors emerged related to individual layers of health inequities, which indicates that the implementation of the industrial mining projects widens the equity gap in local communities. Moreover, our findings suggest that the original layer of “living and working conditions” can be divided into two separate layers, namely “living environmental and subsistence work” and “job opportunities and working conditions in the mine”. Figure 3 also reveals that adding a separate layer for the employment in the mine further illustrates an increase in inequities within affected communities. A recent study observed a similar trend by researching the “resource curse” and “resource blessing” in local communities in areas of oil and gas exploration [57].

### 4.3. Addressing Health Inequities

Addressing health inequities needs holistic health policies, which are acting on different layers and, hence, including the root causes, such as social and economic drivers of health inequities [44,58]. This is in line with the current “Health in All Policies” debate governed by WHO, which emphasizes that health is largely determined outside the health sector [59]. Similarly, the intersectionality theory addresses inequities comprehensively by considering multiple, intertwined factors, including people’s social location, power relations and experiences [60]. Using an intersectionality perspective for investigating health inequities also includes social stratification of populations and, thus, considers different populations groups and subgroups [61]. For instance, considering differential impacts between females and males, including girls and boys, due to their engendered roles is key to effectively address the root causes of health inequities [48]. To address inequities, particularly socially disadvantaged groups must be educated and empowered, as they are likely to be affected most negatively [62]. Strengthening community networks as mitigation strategy for social impacts can empower socially disadvantaged population groups, which usually have a poor social network, and low social status, and, hence, is critical to reduce inequities [44,62]. In this regard, supporting community funds is a promising contribution of the mines, as it allows the communities to address their needs by participating in the decision-making process. As expected, education emerged as an important factor for inequities in our study. It follows that education warrants to be considered more prominently in mitigation strategies.

Although potentially unintended, the communities perceived various impacts especially in contrast of the mine, including implications for their health. Importantly, negative impacts on local communities, but also the unequal distribution of potential benefits, resulted in perceived inequities. This predominately negative perception of surrounding communities is in contrast with recent findings about objective health indicators, indicating positive effects of extractive industries on health outcomes and health determinants. For instance, increase in life expectancy and improved access to drinking water in producer regions were reported. However, such observations should be made available for communities for preventing misperceptions. Thus, community engagement is essential as perceived impacts may be even more important than objectively measured impacts and to address their needs adequately. Overall, improved management of impacts is clearly needed for tackling perceived inequities, especially to “leave no one behind” as promulgated by the 2030 Agenda for Sustainable Development [1,2].

### 4.4. Addressing Health Inequities in the Context of Extractive Industries

Impact assessments are in most countries routinely conducted as part of the licensing process, embracing environmental [63], social [64,65] and, more recently, also health aspects [66]. HIA is a process, which systematically judges the potential, and sometimes unintended, effects of a project on the health of a population and the distribution of those effects within the population [66,67]. Equity—as a guiding principle of HIA—underlines the importance of considering particularly most vulnerable population groups (e.g., women, adolescents, and children) [67,68]. Against this theoretical background, as well as confirmed by few existing case studies [69,70], HIA holds promise to address health inequities in settings of natural resource extraction in LMICs [31].

To unfold the potential of HIA for health equity, the HIA approach has been complemented with specific guidelines for “health equity impact assessment” or “equity-focused health impact assessment” [71,72,73]. To reduce inequities, these guidelines emphasize the need of assessing negative consequence of unintended impacts, considering differential impacts on people or population subgroups and reducing avoidable and unfair factors determining potential inequitable impacts. Another key strategy for integrating equity measures in impact assessment is community empowerment [53,74]. Therefore, researchers have suggested the increased use of participatory assessments in comparative settings and to ensure the inclusion of communities in solution-finding and decision-making processes [75,76,77].

There remains a gap between theory and practice as HIA is currently under-used in LMICs [31,78,79]. In order to maximize the untapped potential of HIA [80], awareness must rise and HIA needs to be strengthened and institutionalized in LMICs [81]. Moreover, HIA must include community members to address their needs adequately. In line with “Health in All Policies” and the call for equity promoting policies [27,82], anchoring community-based HIA in policies should be considered as a first step toward more equitable outcomes [83].

## 5. Limitations

Our study has several limitations. First, the procedures, methods, and analysis of the research undertaken in the three countries with different languages was challenging. The close collaboration of the international team allowed us to collect and analyze the data from the different settings in a standardized manner. Data collection was facilitated by a core research team in each country, who was in charge of training moderators of the FGDs and continuously assuring the quality of the transcription. The varying number of coded references, reflect differences in the data quality and quantity across sites and countries with potential bias for inter-rater reliability. Notably, the analysis was performed by researchers from the core study team, who were familiar with the entire data set and the research context. Regular meetings and exchanges among researchers ensured comprehensive understanding of the data form the different study sites as well as a gradual calibration in the interpretation of the data. The Nvivo project was constantly updated, allowing researchers to learn from each other and harmonize as the coding tree evolved.

Second, the current paper is based on findings retrieved from a suite of FGDs determining perceived health impacts and not health equity per se. Our analysis builds on an initially created node and, hence, country- or site-specific aspects may have remained concealed.

Third, all selected study sites were active mining sites for several years and, hence, not comparable with non-mining sites. It is conceivable that similar patterns can be observed in settings of “natural” urbanization, which is another key issue in the discussion about social determinants of health [25]. However, our study conducted in a host of mining sites in three African countries resulted in consistent evidence of the particular dynamic induced by the project developments. Focusing on perceived inequities, we could yet not assess differences between the mining sites in the current study. This was, however, done in broader papers published under the framing research project [14,43,48].

Fourth, our study is purely reflecting the perspective of affected communities. While it would have been interesting integrating voices from local governments or those operating the mines, this has been addressed in a separate study [84]. Such kind of data triangulation could especially contribute to discuss the roles of governments and mining companies in more detail, which remained limited based on the communities’ perspectives only. To address the potential negative attitude of participants toward the mine, the informed and trained research assistants stated the neutrality of the research at the beginning of the sessions and probed for positive and negative impacts induced by the mine during the FGDs. As we observed similar issues across the sites and countries, perceived inequities are likely to be linked to structural problems.

## 6. Conclusions

The 2030 Agenda for Sustainable Development stipulates “to leave no one behind”. As shown in this study, extractive industries can contribute to the attainment of the SDGs, including the development of surrounding communities; yet negative changes predominate the perception of surrounding communities. With regard to health equity, limited job opportunities and loss of subsistence work caused social instability and increased ill-health among local communities, as they reported. Positioning our findings against the wider determinants of health indicates that the implementation of the mines widened the equity gap. Given the fact that both perceived positive and negative impacts resulted in unequal opportunities for health across settings and countries, there is a pressing need for action to reduce inequities. In order to leave no one behind, impact mitigation must minimize negative consequences of potential unintended impacts and ensure equal opportunities to benefits from positive impacts. Therefore, including a strong equity and community participation component in HIA practice presents an opportunity for addressing the equity gaps identified. Hence, alongside evidence-based policies, the institutionalization of HIA in producer regions is needed to reduce inequities in contexts of natural resource extraction and for striving toward sustainable development in LMICs and beyond.

## Figures and Tables

**Figure 1 ijerph-18-11015-f001:**
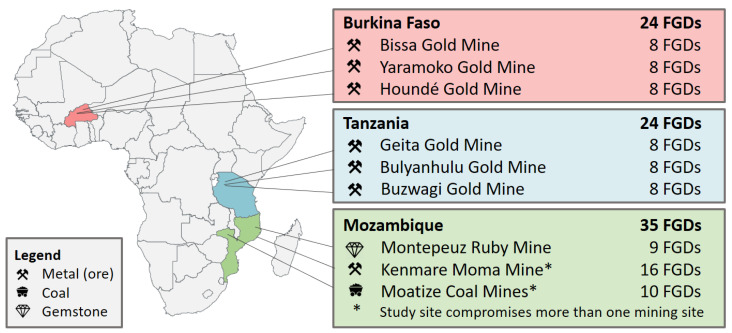
Overview of study sites in three African countries, indicating location, the type of mining project, and the number of focus group discussions (FGDs) conducted.

**Figure 2 ijerph-18-11015-f002:**
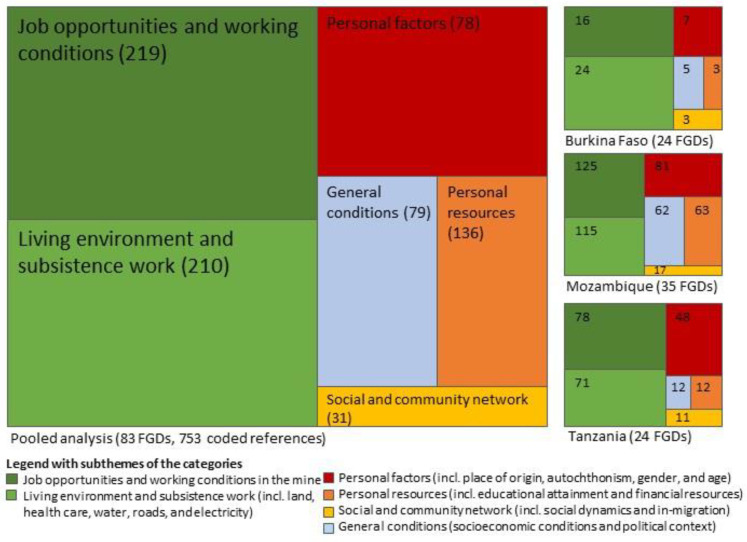
Qualitative tree maps indicating the proportion of coded references (and absolute number in brackets) of perceived inequities and the consistency of findings across countries (FGD: focus group discussion).

**Figure 3 ijerph-18-11015-f003:**
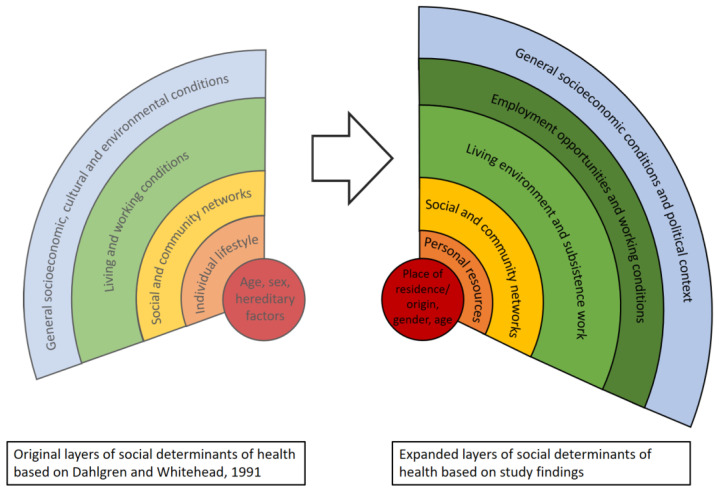
Layers of social determinants of health. Contrasting layers of inequities from the original model (Dahlgren and Whitehead, 1991; **left**) and expanded model based on the perception of communities affected by industrial mining projects (our own data; **right**).

**Table 1 ijerph-18-11015-t001:** Characteristics of study participants. Number of study participants and sociodemographic characteristics by country and in total (FGD: focus group discussion).

	Burkina Faso (24 FGDs)	Mozambique (35 FGDs)	Tanzania (24 FGDs)	Total (83 FGDs)
**Number of study participants (and relative frequency in %)**				
Male	115 (49.8%)	181 (48.0%)	89 (48.6%)	385 (48.7%)
Female	116 (50.2%)	196 (52.0%)	94 (51.4%)	406 (51.3%)
Total	231	377	183	791
**Average number of participants per FGD (and range)**				
Male	10 (6–10)	11 (6–12)	8 (6–8)	9 (6–12)
Female	10 (7–10)	11 (8–13)	8 (6–10)	9 (6–13)
**Total**	**10 (6–10)**	**11 (6–13)**	**8 (6–10)**	**9 (6–13)**
**Average age in years (and age range)**				
Male	42 (23–71)	45 (19–89)	48 (19–77)	45 (19–89)
Female	31 (18–49)	44 (29–83)	42 (20–77)	39 (18–83)
Total	37 (18–71)	44 (19–89)	45 (19–77)	42 (18–89)
**Average years living in the community (and range)**				
Male	26 (3–67)	37 (3–89)	22 (2–66)	30 (2–89)
Female	13 (1–44)	36 (1–83)	19 (1–77)	25 (1–83)
Total	20 (1–67)	37 (1–89)	21 (1–77)	28 (1–89)
**Average number of years of school attended (and range)**				
Male	2.7 ^1^ (0–10)	3.8 (0–12)	7.4 (0–14)	4.4 (0–14)
Female	1.2 ^1^ (0–10)	1.4 (0–12)	7.3 (0–14)	2.9 (0–14)
Total	1.9 ^1^ (0–10)	2.6 (0–12)	7.4 (0–14)	3.6 (0–14)

^1^ Data from two FGDs with male participants and two FGDs with female participants missing.

## Data Availability

The data used for this paper are of qualitative nature (i.e., verbatim transcript of focus group discussions). Participants shared personal information, which is difficult to anonymize completely. Further, participants have not provided informed consent for their data to be stored in a public repository. Given these reasons and to guarantee confidentiality, we would rather not share our data in a public repository. Requests to access the datasets should be directed to the corresponding author.

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
