# Peer review of "“It Is Like We Are Living in a Different World”*: Health Inequity in Communities Surrounding Industrial Mining Sites in Burkina Faso, Mozambique, and Tanzania"

_ijerph, 2021, doi:10.3390/ijerph182111015_

Round 1

Reviewer 1 Report

Abstract

  • Line 22: needs a brief explanation of the framework method.
  • Line 29: “To strive toward a more equitable world” this sentence has a wide scope. It needs to be better linked to the aims of the study, as provided in the abstract

Introduction

General comment: Well organised and written. The background can be strengthened with additional support on relevant literature. Some references suggested.

Specific comments:

  • Lines 53-54. Stronger background needed. Consider these references: DOI: 10.1016/j.exis.2015.08.006 and 10.1186/s12889-018-5505-7
  • Lines 57-58. Is this question being portrayed as a research question of the study?
  • Line 72. Better explanation needed for sentence “structural drivers of inequities, such as power”
  • Lines 82-87. Stronger support for the incorporation of studies that address mining impacts on at-risk communities is needed. Consider reviewing https://doi.org/10.1515/reveh-2019-0033 and   DOI:  1186/1476-069X-7-61
  • Lines 90-91. The second research question needs further clarification

Methods

General comments: Was a comparison between different types of mining (gold, coal, et…) considered? This would add robustness to the analysis. Some sections need further clarification (see comments below):

Specific comments:

  • Line 115. Were other geographical aspects of proximity considered for the selection of communities? Also, some review of their socioeconomic background?
  • Line 116. What was considered positive/negative impacts?
  • Lines 141-152. There is potential confusion on the methods used (there are mentions to “the framework method”, “the SDH framework” and “the WHO framework”.

Results

General comment: strongly suggest including a table with the summary of findings by themes or broad issues identified.

Specific comments:

  • Line 258. “the original framework”: Not clear what do you refer to?
  • In line 430 you addressed the consistency of findings across countries. It would add strength to the analysis to analyse the consistency of findings across types of mining (linked to a comment in the methods section).

Discussion

  • Section 4.4. should consider other impact assessment methods (SIA and some components of EIA)

Limitations

  • potential bias for inter-rater 564 reliability was mentioned. How did this affect the analysis and how was managed?
  • How the differences in types of mining could have affected the uniformity of conclusions?

Conclusions

  • Lines 583-588 could be better placed at the start of the discussion.
  • Lines 590-602 need to be better linked to the aims/research questions of the study

Author Response

Please see attachment. Changes are highlighted using a yellow marker and line numbers refer to the marked-up version of the manuscript. 

Reviewer 2 Report

This is a qualitative study, and as authors have claimed, the results come from 83 focus group discussions, enrolling 791 participants from 10 study sites in Burkina Faso, Mozambique, and Tanzania.  The FGDS were audio-recorded, and transcripts have analyzed using Nvivo software.

This is very interesting and important study but After reading this manuscript, I have a few major issues:

1)  Authors mentioned that “All FGDs were audio-recorded and transcribed verbatim into French (Burkina 132 Faso), Portuguese (Mozambique), or English (Tanzania)”, but the coding happened by using in-equality in English? Would you please clarify if the authors have converted all FG transcripts into EN for analysis?

2) The study happened in three countries, but the results reported as one study, it is tough to believe there were no differences between countries.

3) The authors negatively highlighted almost any kind of activities performed by ‘the mines companies’. However, the authors mentioned a few positive impacts such as” providing temporary job opportunities, improving road, etc., but overall the manuscript approach is so NEGATIVE. It may be because of several issues:

Sample bias: As stated in the MS, they randomly selected people >18 for FG discussion. However, the study has not mentioned if they had held any focus groups with:

  1. people who companies have hired or community leaders who had positions with the ‘mining companies’
  2. ‘policy makers’ for the local government to discuss the positive sides of the mining companies' activities..

Bias in Results: This type of bias backs to authors or study teams; there are huge arguments around inequality, but the bottom line is all types of inequality is not bad things, for example, look at Marmot publications on income inequality and health or and health or publications by Angus Deaton, etc.

As reading the text, there is confusion with the concept of economic inequality. For example, on page 7: "to be community leader you need to pay someone, now people are fighting to be community leader, those who had no decent house have built it […]. Those who never had a car now have a car. So they are fighting to be community representatives, nowadays people have already opened their eyes, no one is robbed only the farmers who go to the fields all the time".

The sentence raised a question about corruption but also provided evidence the "mine project' improved the communities by hiring people, pay them enough to build a house, buy a car, etc. Of course, this is inequality, but also opportunity. The point is that the mission of 'mining companies' – more likely private companies – is different from the government's mission to establish an environment to reduce inequality.

  1. Still, we need to look deeply at the role of ‘law and regulation’ or any authorities who permitted the mining project activities. It seems that FG participants by default considered the ‘mine companies’ as government agents to bring job opportunities, hire men and women, and provide healthy and safe water. There is a short sentence in the MS that “The regulations require that a certain share of the companies’ revenue must be invested in community development projects.”, this is the critical point of the paper that needs to be highlighted and discussed well.

Sometimes the positive activities of companies were reported negatively. For example, ‘renovate a road” or ‘creating a road’ is a ‘public good’ and benefits all populations. For example: “they use their own money to renovate a road from [the town] to [the mine] because they use that road to transport most of their commodities”.

There is a mixed-up discussion between corruption, mistrust of government activities, and people here, the participants on the FG for example: “We have no power over them; on the other hand, they have the power to do what they want because the authorities let them do.”

I also have one minor comment:  Page 5 “Average number (and range) of participants per FGD” please round the number; using decimal for the human body does not make sense.

Here are my suggestions:

I have two suggestions:

First, please add some arguments on different approaches to inequality and development and some short and long-term effects, for example, distribution of poverty in not equality.

Second, the discussion needs to improve sufficiently to discuss the elements I have raised above and focus on the people's expectations from the government and the ‘mining project’.

Author Response

(The authors gave the same response as above.)

Reviewer 3 Report

Dear authors, the article is very interesting.

I have minor revisions for you:

2.3 Recruitment and study population 

Please, could you specify the total number of the members of the different communities?

Could you explain why the people under 18 were excluded from the study?

Please, could you set the fields of the table differently? It is not clear enough now.

How it is possible to associate a single quote with the differrent areas, i.e. Burkina Faso, Mozambique, Tanzania? It is not clear for me now....

Author Response

(The authors gave the same response as above.)

Round 2

Reviewer 2 Report

Thank you for addressing my comments, no further comments.